# On the reproducibility of science: unique identification of research resources in the biomedical literature

Nicole A. Vasilevsky[1], Matthew H. Brush[1], Holly Paddock[2],
Laura Ponting[3], Shreejoy J. Tripathy[4], Gregory M. LaRocca[4] and
Melissa A. Haendel[1]

[1] Ontology Development Group, Library, Oregon Health & Science University, Portland, OR, USA
[2] Zebrafish Information Framework, University of Oregon, Eugene, OR, USA
[3] FlyBase, Department of Genetics, University of Cambridge, Cambridge, UK
[4] Department of Biological Sciences and Center for the Neural Basis of Cognition, Carnegie Mellon University, Pittsburgh, PA, USA

Corresponding author
Nicole A. Vasilevsky,
vasilevs@ohsu.edu

## ABSTRACT

Scientific reproducibility has been at the forefront of many news stories and there exist numerous initiatives to help address this problem. We posit that a contributor is simply a lack of specificity that is required to enable adequate research reproducibility. In particular, the inability to uniquely identify research resources, such as antibodies and model organisms, makes it difficult or impossible to reproduce experiments even where the science is otherwise sound. In order to better understand the magnitude of this problem, we designed an experiment to ascertain the "identifiability" of research resources in the biomedical literature. We evaluated recent journal articles in the fields of Neuroscience, Developmental Biology, Immunology, Cell and Molecular Biology and General Biology, selected randomly based on a diversity of impact factors for the journals, publishers, and experimental method reporting guidelines. We attempted to uniquely identify model organisms (mouse, rat, zebrafish, worm, fly and yeast), antibodies, knockdown reagents (morpholinos or RNAi), constructs, and cell lines. Specific criteria were developed to determine if a resource was uniquely identifiable, and included examining relevant repositories (such as model organism databases, and the Antibody Registry), as well as vendor sites. The results of this experiment show that 54% of resources are not uniquely identifiable in publications, regardless of domain, journal impact factor, or reporting requirements. For example, in many cases the organism strain in which the experiment was performed or antibody that was used could not be identified. Our results show that identifiability is a serious problem for reproducibility. Based on these results, we provide recommendations to authors, reviewers, journal editors, vendors, and publishers. Scientific efficiency and reproducibility depend upon a research-wide improvement of this substantial problem in science today.

## INTRODUCTION

The scientific method relies on the ability of scientists to reproduce and build upon each other's published results. Although it follows that the prevailing publication model should support this objective, it is becoming increasingly apparent that it falls short (*Haendel, Vasilevsky & Wirz, 2012*; *de Waard, 2010*). This failure was highlighted in a recent Nature report from researchers at the Amgen corporation, who found that only 11% of the academic research in the literature was reproducible by their groups (*Begley & Ellis, 2012*). Further alarm is raised by the fact that retraction rates, due in large part to a lack of reproducibility, have steadily increased since the first paper was retracted in 1977 (*Cokol, Ozbay & Rodriguez-Esteban, 2008*). While many factors are likely at play here, perhaps the most basic requirement for reproducibility holds that the materials reported in a study can be uniquely identified and obtained, such that experiments can be reproduced as faithfully as possible. Here, we refer to reproducibility defined as the "conditions where test results are obtained with the same method on identical test materials in different laboratories with different operators using different equipment" (*ISO 5725-1:1994, 1994*). This information is meant to be documented in the 'materials and methods' of journal articles, but as many can attest, the information provided there is often not adequate for this task. Such a fundamental shortcoming costs time and resources, and prevents efficient turns of the research cycle whereby research findings are validated and extended toward new discoveries. It also prevents us from retrospectively tagging a resource as problematic or insufficient, should the research process reveal issues with a particular resource.

Until recently, challenges in resource identification and methodological reporting have been largely anecdotal, but several efforts have begun to characterize this problem and enact solutions. The National Centre for the Replacement, Refinement and Reduction of Animals in Research (NC3R) evaluated methodological reporting in the literature for *in vivo* studies using rodent models or non-human primates. They examined 271 publications and reported that only 60% of the articles included information about the number and characteristics of the animals (strain, sex, age, weight) and approximately 30% of the articles lacked detailed descriptions of the statistical analyses used (*Kilkenny et al., 2009*). Based on this study, the ARRIVE guidelines (http://www.nc3rs.org.uk/page.asp?id=1357) were developed for reporting of *in vivo* experiments pertaining to animal research. Other domain specific standards have been published such as the Minimum information about a protein affinity reagent (MIAPAR) (*Bourbeillon et al., 2010*) and the high-profile communication from Nature to address concerns regarding research reproducibility where they offered improved standards for reporting life science research (http://www.nature.com/authors/policies/reporting.pdf). The Neuroscience Information Framework (NIF; http://neuinfo.org) specifically developed the Antibody Registry as a means to aid identification of antibodies within published studies, based on a small pilot study which showed that >50% of antibodies could not be identified conclusively within published papers (AE Bandrowski, NA Vasilevsky, MH Brush, MA Haendel, V Astakhov, P Ciccarese, J McMurry and ME Martone, unpublished data). ISA-TAB provides a generic, tabular format, which contains metadata standards to facilitate data collection, management,

and reuse (*Sansone et al., 2012*; *Sansone, 2013*; *Thomas et al., 2013*). To promote scientific reproducibility, the Force11 community has published a set of recommendations for minimal data standards for biomedical research (*Martone et al., 2012*) and published a manifesto to improve research communication (*Phil et al., 2011*). The BioSharing initiative (www.biosharing.org) contains a large registry of community standards for structuring and curating datasets and has made significant strides towards the standardization of data via its multiple partnerships with journals and other organizations.

While the work highlighted above has offered guidance based on the perceived problem of inadequate methodological reporting, the fundamental issue of material resource identification has yet to be specifically characterized using a rigorous scientific approach. It is our belief that unless researchers can access the specific research materials used in published research, they will continue to struggle to accurately replicate and extend the findings of their peers. Until our long held assumptions about a lack of unique identifiability of resources are confirmed with quantitative data, this problem is unlikely to pique the interest of funding agencies, vendors, publishers, and journals, who are in a position to facilitate reform. To this end, we report here an experiment to quantify the extent to which material resources reported in the biomedical literature can be uniquely identified. We evaluated 238 journal articles from five biomedical research sub-disciplines, including Neuroscience, Developmental Biology, Immunology, Cell and Molecular Biology, and General Biology. Target journals were selected from each category to include a representative variety of publishers, impact factors, and stringencies with respect to materials and methods reporting guidelines. In each article, we tracked reporting of five types of resources: (1) model organisms (mouse, rat, zebrafish, worm, fly, frog, and yeast); (2) antibodies; (3) knockdown reagents (morpholinos or RNAi); (4) DNA constructs; and (5) cell lines. We developed a detailed set of evaluation criteria for each resource type and applied them to determine the identifiability of over 1,700 individual resources referenced in our corpus. The results of this experiment quantify a profound lack of unique identification of research resources in the biomedical literature across disciplines and resource types. Based on these results and the insights gained in performing this experiment, we provide recommendations for how research resource identification can be improved by implementing simple but effective solutions throughout the scientific communication cycle.

## METHODS

### Journal selection and classification

The core of our evaluated corpus was comprised of articles from a set of target journals that varied across three features: research discipline, impact factor, and reporting guideline requirements. For research discipline selection, we followed the Institute for Scientific Information (ISI) categorization and selected five journals from Cell Biology, Developmental Biology, Immunology, and Neuroscience. In addition, a non-ISI category (General Biology) was included to cover multidisciplinary journals such as Science, Nature, and PLoS Biology. Within each discipline, care was taken to include journals with a range of impact factors as reported in the Journal Citation Report from 2011 (*Thomson Reuters, 2011*).

Journals were binned into three categories (high, mid, and low) based on whether their impact factor fell into the top, middle, or lowest third for their discipline in this report. Finally, we selected journals that varied in the stringency of their recommendations for reporting data about material resources. Journals were assigned to one of three categories: (1) *Stringent* if the journal required detailed information or specific identifiers to reference materials reported in the manuscript (e.g., required catalog numbers for antibodies); (2) *Satisfactory* if the journal provided only limited recommendations for structured reporting or resource identifiers, but did not restrict space allocated for this information; and (3) *Loose* where minimal or no reporting requirements for materials and methods were provided, and/or the length of material reporting space was restricted. Note that these guidelines were the ones in effect at the time of manuscript selection (January 18, 2013).

## Article selection

Articles in the core collection of our corpus were selected randomly by performing a PubMed search filtered for each journal and using the first five publications returned on January 18, 2013 (all publications were from 2012–2013). This approach was adequate for all journals except Nature and Science, which cover a very general scientific spectrum such that top PubMed hits often failed to include the resource types evaluated in our study. For these journals, the most recent articles that were likely to contain our resources were selected directly from the publisher's website. Recent publications were chosen for our corpus deliberately to reduce the chance that they had been curated by a model organism database (MOD) or other curatorial efforts, which could skew results by providing additional curated data not reported or accessible from the original article alone. NIF had also noted in a pilot project that the identifiability of reagents decreases over time, as commercial vendors eliminate products from their catalogs.

In addition to this core collection of 135 core articles, we added 86 additional publications to our study through a collaboration with the Zebrafish Information Network (ZFIN), who agreed to assess identifiability of reported resources according to our evaluation guidelines as part of their established curation pipeline. Finally, a set of 17 more articles from the Nathan Urban Laboratory at Carnegie Mellon University was included in our experiment. The Urban Lab studies cellular and systems neuroscience, and extensively uses animal models and antibodies. These articles were included to explore how the thorough and structured documentation practices of this lab in its internal management of resource inventory and usage is reflected in its reporting of materials in the literature they produce. Articles from these additional ZFIN and Urban Lab collections were also classified according to discipline and impact factor, so as to be included with our core collection in our factor analysis. In total, 238 manuscripts were analyzed from 84 journals. All of the articles contained at least one or more of the research resources we evaluated in this study. To ensure this was a sufficient number of papers, we did preliminary statistical analysis to determine that we could find statistical significance in the results. A list of the journals, domains, impact factors, and PubMed IDs, as well as the complete dataset is available in Table S1.

### Article curation workflow

A team of three curators evaluated a selection of articles from the corpus, with each being reviewed by a single expert to identify and establish the identifiability of each documented resource. In addition, zebrafish and fly genetics experts curated the zebrafish and drosophila model organisms, respectively, as our primary curators did not have expertise in these areas. We performed spot-checking of the primary curation and issues found by the secondary evaluator were documented in the curation spreadsheets and updates were made to the curation guidelines. Where necessary, the curator used supplemental data and any referenced articles or publically accessible online data sources, dating as far back as necessary to find uniquely identifying information about a resource. This included vendor catalogs and a variety of experimental and resource databases, where identifying information was often resolvable based on information provided in a publication. More detailed evaluation criteria for unique identification of each resource type are described below. For a given article, evaluation of only the first five resources of each type was performed in the core publication collection. This was necessary as some papers referenced a cumbersome number of resources such as antibodies or RNAi oligos, which were typically reported to the same degree of rigor.

### Resource identification criteria

Based on our extensive experience in working with these particular resources and on consultation with several external experts, we developed a set of criteria to determine the ability of each resource type to be 'uniquely identified'. Generally, 'unique identification' requires that a specific resource can be obtained or created based on information provided in or resolvable from the publication directly, or resolvable through referenced literature, databases, or vendor sites. Below we outline some general and resource-type specific requirements for 'identifiability' applied in our evaluations.

## GENERAL CONSIDERATIONS

### Catalog numbers

For commercial resources, provision of a catalog number and the name of the vendor that resolves to a single offering uniquely identifies a resource. In the absence of a catalog number, if provision of only the vendor and resource name allows unambiguous resolution to a single offering, a resource is considered identifiable. For example, reporting "polyclonal anti-HDAC4 from Santa Cruz" resolves to a single antibody in the Santa Cruz catalog even without a catalog number. However, this is not ideal, because the catalog may expand to include additional polyclonal anti-HDAC4 antibodies in the future, which would render the resource unidentifiable. Additionally, catalog numbers are not stable as products are discontinued or sold; hence we also looked for a record of the antibody in the Antibody Registry (www.antibodyregistry.org), which provides stable IDs for antibody offers.

## Sequence molecule identification

Sequence identification is a central aspect of identifiability for many resource types. Examples include specifying the sequence of an immunogenic peptide for a lab-sourced antibody, the sequence of a DNA insert of a construct, or the sequence of a transgene incorporated into the genome of an organism or cell line. In such cases, these sequences need to be resolvable to known information about the specific nucleic acid or peptide sequence to support identifiability of the resource to which they are related. Criteria that establish resolution of a sequence in support of identifying a dependent resource include: (1) directly providing the full sequence; (2) referencing a resource from which the sequence can be determined (to the extent that it is known)—e.g., by providing a gene ID or accession number that can be looked up and a sequence determined; (3) when precise/complete sequence information does not exist, a sequence should be tied to some other unique entity, such as a single, unique source and procedure through which the physical sequence can be obtained/replicated (e.g., primers and a specific source of template DNA such as a uniquely identified cell type or biological sample). The requirement for complete resolution to a specific sequence is not absolute as it is sometimes the case that this information is not known, and for some resource types a complete sequence may not be required to be considered uniquely identifiable. One recurring theme we encountered in our study was authors referencing a gene name or sequence to identify cDNA or a peptide related to the gene. This can be problematic, as specification of a gene sequence may not be sufficient to resolve a single cDNA or peptide sequence. This is because a single gene may resolve to many different transcripts or peptides (e.g., through alternative splicing), which can prevent unambiguous resolution of a gene sequence to a cDNA or peptide sequence.

# SPECIFIC RESOURCE IDENTIFICATION CRITERIA

## Antibodies

Unique antibody identification required at least one of the following: (1) an identifier resolving to a universal registry/database identifier such as the Antibody Registry (www.antibodyregistry.org) or eagle-i repository (http://www.eagle-i.net), or a vendor name and catalog number for resolving to a single offering; (2) for antibodies not publicly available, sufficient protocol details on production of the antibody so as to allow reproduction. This detail minimally includes specifying the host organism and identity of the immunogen used. For peptide immunogens, criteria for sequence identification above apply, i.e., that an immunogenic protein or peptide resolves to single gene product sequence. Note that the criteria for identifiability do not include the lot or batch number, although a case could be made for this level of granularity.

## Organisms

For 'wild-type' organism strains, an unambiguous name or identifier, such as a stock number, the official International Mouse Strain Resource (IMSR) name or a MOD number, is required as well as a source vendor, repository, or lab. For genetically modified

strains, identifiability requires reporting or reference to all genotype information known, including genetic background and breeding information, and precise alterations identified in or introduced into the genome (including known sequence, genomic location, and zygosity of alterations). For random transgene insertions, it is not required that genomic location of insertion(s) is known, but precise sequence of inserted sequence should be unambiguously resolvable according to sequence identification criteria above. For targeted alterations, genomic context of the targeted locus and the precise alterations to the locus should be specified according to sequence identification criteria above. This information can be provided directly, or through reference to a MOD record or catalog offering where such information is available. The MODs provide specific nomenclature guidelines that are consistent with these views.

## Cell lines

For standard publically available lines, an unambiguous name or identifier is required as well as a source for the line (e.g., a vendor or repository). This information should resolve to data about the organismal source and line establishment procedures. For example, a common cell line reported that can be obtained from ATCC would be considered identifiable, however, if only the name of the line is mentioned without any other identifying information then it is considered unidentifiable. For novel lab-generated cell lines, an organismal source (species and known genotype information, anatomical entity of origin, developmental stage of origin) and any relevant procedures applied to establish a stable lineage of cells. Additionally, some indication of passage number is recommended but not strictly required. For genetically modified lines, identifiability criteria are analogous to those for genetically modified organisms, including genomic location and zygosity or copy number of modifications where this information is known.

## Constructs

Construct backbone should be unambiguously identified and resolvable to a complete vector sequence (typically through a vendor or repository). The sequence of construct inserts should be identifiable according to sequence identification criteria above. Most expression constructs incorporate cDNA—so it is particularly important that the exons included in this insert are resolvable when more than one splice variant exists for a gene transcript. This means that specifying the name of a gene or a protein expressed may not be sufficient if this does not allow for unambiguous resolution to a cDNA sequence. Identification does not require precise description of MCS restriction sites used for cloning, but this information is encouraged. Relative location and sequence of epitope tags and regulatory sequences (promoters, enhancers, etc.) should be specified (e.g., 'N-terminal dual FLAG tag' is sufficient). For example, referencing the accession number and the vector backbone is sufficient to identify the construct, as in: "for the full-length Dichaete construct, the insert was amplified from the full-length cDNA clone (GenBank accession X96419 and cloned into the HindIII and KpnI sites of pBluescript II KS(!)" (*Shen, Aleksic & Russell, 2013*). However, in most constructs, such level of detail is omitted.

### Knockdown reagents

Identifiability requires specific and complete sequence identification according to the criteria outlined above. This will typically be direct reporting of the sequence, as these are generally short oligos. For example, this text provided in the method section was considered identifiable: "The DNA target sequence for the rat Egr-2 (NM_053633.1) gene was CAGGAUCCUUCAGCAUUCUTT" (*Yan et al., 2013*). In cases where sequence information was not provided, the reagent was considered unidentifiable.

### Statistical analysis

Since the data was binomial in that each resource was either identifiable or not, we used a binomial confidence interval strategy for calculating upper and lower 95% confidence intervals (CI) (http://www.biyee.net/data-solution/resources/binomial-confidence-interval-calculator.aspx). Error bars for the corresponding 95% CI are displayed on the graphs. Statistical significance was determined by calculating the z-score.

## RESULTS AND DISCUSSION

The goal of our study was to determine the proportion of research resources of five common types that can be uniquely identified as reported in the literature. 'Unique identification' requires that a resource can be obtained or re-created based on information provided in or resolvable from a publication. The criteria for identifiability were established a reasonable level of granularity, recognizing that finer levels, e.g., lot or litter number, may be possible. Establishing identifiability criteria was central to our effort, and these criteria are complex and varied between resource types as described in the Methods section. The results of our study provide quantification of this problem in the literature. In total, only 54% (922/1703) of evaluated resources were uniquely identifiable. Considerable variability was found across resource types (Fig. 1A), which may result from the inherent differences in the attributes relevant to their identification, or from the level of external support for applying identifiers and metadata for their unique identification. In addition, the level of identifiability for each resource type is tied directly to the stringency of the criteria that were separately developed for each, which are unavoidably exposed to some degree of subjectivity.

### Antibodies

Antibody reagents represent one of the most challenging and important resource types to adequately identify, given their ubiquitous use, expense to create, and condition-specific efficacy. The most common issue with reporting of antibodies was a lack of catalog number (for commercial antibodies) or a lack of reference to the immunogen used to generate the antibody (for non-commercial antibodies). A separate analysis of commercial versus non-commercial (e.g., lab-made) antibodies showed an average of 46% of commercial antibodies, and similarly, 43% of non-commercial antibodies were identifiable. While commercial suppliers do an acceptable job of providing basic metadata about their offerings (for example, see http://datasheets.scbt.com/sc-546.pdf), the market is flooded with products of variable quality metadata. In practice, the literature is where most

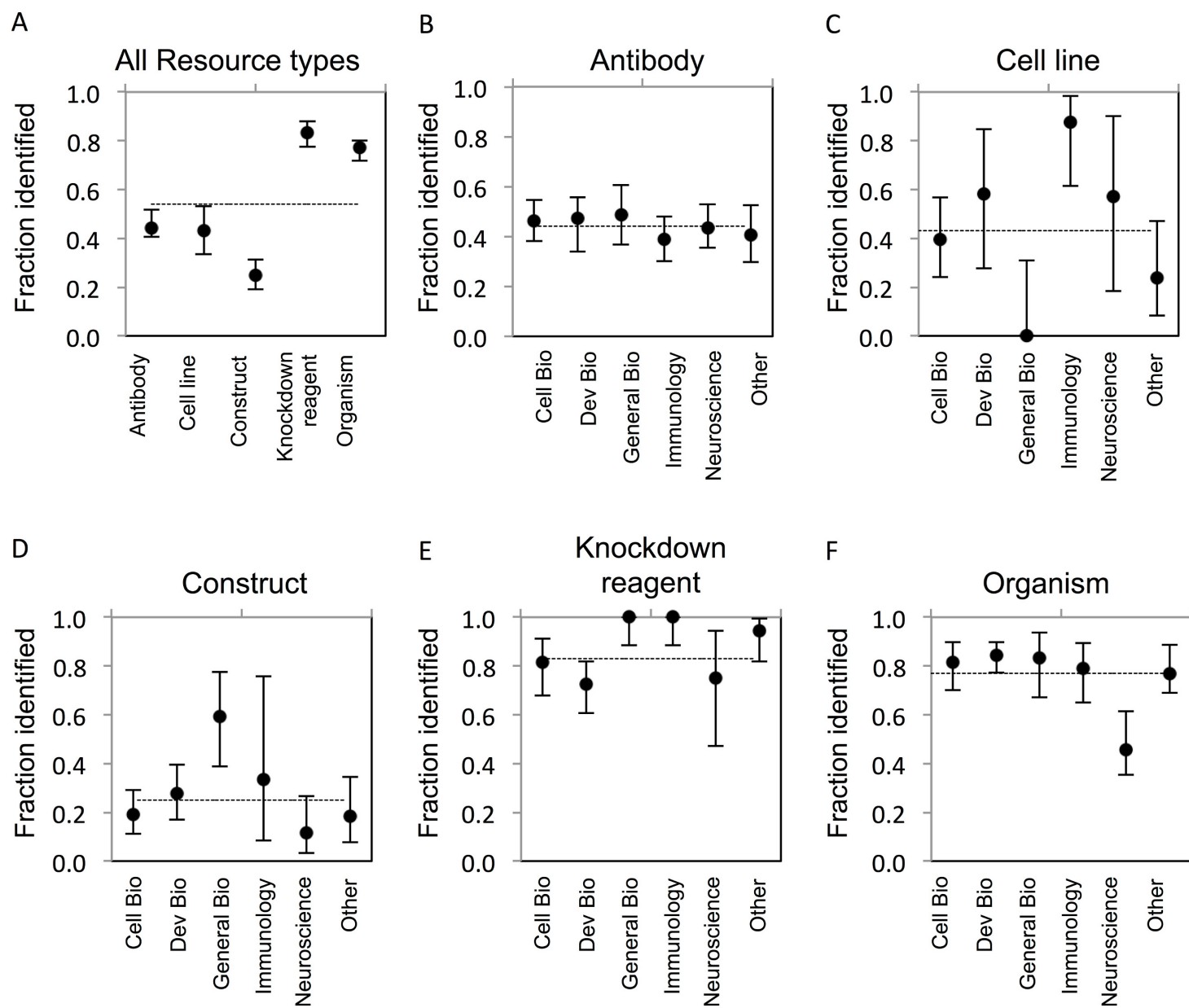

**Figure 1 Resource identifiability across disciplines.** (A) Summary of average fraction identified for each resource type. (B–F) Identifiability of each resource type by discipline. The total number of resources for each type is: (B) antibodies, $n = 703$; (C) cell lines, $n = 104$; (D) constructs, $n = 258$; (E) knockdown reagents, $n = 210$; (F) organisms, $n = 428$. The $y$-axis is the average for each resource type across each domain. Variation from this average is shown by the bars, error bars indicate upper and lower 95% confidence intervals.

scientists look when searching for the right antibody for their work, as evidenced by a marketing report from 1 Degree Bio (http://1degreebio.org/) showing 63% of researchers use journal references to guide antibody selection (A Hodgson, unpublished data). This makes it all the more troubling that only 44% of antibodies evaluated in our study could be uniquely identified (Fig. 1B). While reporting of a catalog number alone is considered

sufficient for unique identification of a commercially available antibody, we found they were provided for only 27% of antibodies we evaluated.

A likely reason for the shortcoming in commercial antibody identification may be that journal reporting guidelines rarely require catalog numbers be reported for antibodies (or any other reagent type for that matter). More commonly, only a name and location of a manufacturer are required. For example, the journal "Immunology" simply states: "Materials and Methods: sufficient information must be included to permit repetition of experimental work. For specialist equipment and materials the manufacturer (and if possible their location) should be stated" (*Wiley Online Publishing*). By contrast, the Journal of Comparative Neurology (JCN) is one of the rare journals that do require more precise reporting of antibody metadata, including their catalog number. An extensive evaluation of 6,510 antibodies in the JCN Antibody Database (*Wiley Online Publishing, 2013*) revealed that a catalog number was reported in over 90% of the antibodies captured in their database (AE Bandrowski, NA Vasilevsky, MH Brush, MA Haendel, V Astakhov, P Ciccarese, J McMurry and ME Martone, unpublished data, and re-evaluated in this study). This highlights how a simple solution such as requiring catalog number reporting can vastly improve resource identification in the literature.

Notably, as more data is becoming available about protein structure, localization, and function, the identity of peptide immunogens and epitopes used in creating an antibody becomes increasingly valuable for explaining its performance in different applications. Identification and tracking of immunogens is one area where there is considerable room for improvement among vendors and resource databases. Efforts such as the Immune Epitope Database (IEDB) (http://www.iedb.org/), a manually curated repository of immunological data about epitope recognition, can be looked to for guidance in how to capture and represent relevant data about such epitopes. The IEDB curates papers that report discovery of new epitopes and even in this very specific use case where the goal is to report on the specific epitope, only 81% of the epitopes they curated had the epitope sequence provided in the published manuscript (R Vita, unpublished data).

## Cell lines

A source for cell lines was rarely reported and the lack of source was the most common factor for their low identifiability in our study. For commonly used, unmodified lines such as HEK293T cells, our guidelines required a source be provided in addition to the line name. This information was deemed important given the tendency of lines propagated in isolation to diverge genetically through continuous passages (*Hughes et al., 2007*). There are increasingly documented occurrences of cell line misidentification and contamination, as highlighted by the infamous HeLa contamination statistics (*Gartler, 1968*) and other cell line contaminations (*Phuchareon et al., 2009*). Simply reporting the name of the line without a source fails to provide any information on the history and integrity of the line. For lab-generated or genetically modified cell lines not available from a public source, identification required a basic description of the line's establishment procedure, its anatomical source, and/or the precise genetic modifications made (see details in Methods section).

Based on these criteria, the identifiability of cell lines was comparable to that for antibodies, averaging 43% across all disciplines (Fig. 1C). A notable difference was found for cell line identifiability between our lowest and highest reporting disciplines—General Biology (0% identifiable) and Immunology (88% identifiable). This may reflect the tendency for less rigorous reporting requirements and reduced space allocation for methods that are common in high-profile journals we included in this category (e.g., Nature, Science). By contrast, the majority of cell lines reported in Immunology papers adequately referenced either the lab, investigator, or commercial supplier that provided the cell line, which may indicate more rigorous conventions for sharing and attribution for cell lines in this community; however, due to the low number of cell lines evaluated in immunology journals in this study, we cannot make this conclusion.

An important aspect of cell lines that we found highly neglected in literature reporting was passage number. This attribute provides an important metric to gauge the integrity of a cell line sample, and how likely it is to be faithfully reflected in another sample. We found such information to be rarely reported in our study, and thus did not require it in addition to a source for identifiability. But we highly recommend more attention be paid to tracking and reporting this important attribute in the literature. This practice is particularly important for lines propagated in research labs, as a survey on cell line usage reported that 35% of researchers use cell lines obtained from another lab rather than a cell line repository (*Buehring, Eby & Eby*). Tracking passage number and contamination is a lower priority in these labs compared to commercial repositories, such that the use of genetically or compositionally divergent samples of the same line is likely to be a significant contributor to difficulties in reproducing cell-line based research. Towards this end, a guideline has been published to check for contamination and authenticity of cell lines (*Capes-Davis et al., 2010*).

## DNA constructs

Unique identification of constructs was the lowest amongst all resource types examined, on average 25% were identifiable, due to lack of reporting of sequence or other identifying information (Fig. 1D). This was likely due to the dependency of identification on reporting a complete or approximated sequence, and the lack of incentive, guidelines, or technical support for providing such metadata. While many construct backbones are obtained from commercial manufacturers where the relevant sequence information is provided, the valuable component of a construct are the gene(s) that have been sub-cloned in by a researcher. Access to this sequence information is critical in order to reproduce the experiment or fully utilize these resources, but it is rarely directly reported in full. While resources like Addgene and PlasmID provide detailed information about constructs and the relevant gene components, submission of plasmids to such repositories is infrequent, as we found less than 10% of non-commercial plasmids reported in our corpus to be present in such repositories. In cases where primer sets were used to generate a construct insert, we often found that the primer sequences were reported; yet the specific and complete

sequence of the amplified template was rarely specified. In such cases, it is not possible to determine the sequence of the product cloned into a construct.

## Gene knockdown reagents

Knockdown reagents were significantly more identifiable compared to the former resource types mentioned above, at 83% (Fig. 1E). Knockdown reagents are frequently used, in particular in Cell and Developmental Biology (*Harborth et al., 2001*; *Nasevicius & Ekker, 2000*). Identifiability of knockdown reagents was the highest amongst resource types. This is likely due to the fact that knockdown reagents tend to be comprised of short, and therefore easy to include, sequence information. Additionally, editors often require reporting of sequences for custom reagents, as this information is critical to understanding and verifying the reagent function. MODs also keep track of these sequences as they curate papers. The majority of knockdown reagents that were curated in this study were from Developmental Biology journals, which also had the lowest number of identifiable reagents compared to other fields. Knowing the exact sequence used is necessary to reproduce the experiment, and concentration and experimental details are similarly important to determine off-target effects.

## Organisms

Organisms showed a relatively high identifiability of 77% (Fig. 1F). Amongst organisms, yeast were the most identifiable (100%, albeit there were only 5 strains analyzed from one paper), followed by zebrafish (87%), flies (80%), mice (67%), and rats (60%). Worms and frogs were the least identifiable, at 58%, 33%, and 0%, respectively. The identification of transgenic organisms was higher, with 83% of transgenic organisms being identifiable compared to 46% of non-transgenic wild type strains. The higher identifiability may be due to the fact that 56% of the transgenic strains we analyzed had already been curated by a MOD, because the organisms reported in our corpus were previously reported in an earlier publication that had been curated by a MOD. Indeed, identifiability of organisms not found in a MOD was considerably lower at 60%. The MODs review the current literature and annotate information about genetic modifications used in transgenic strains, phenotypes, gene expression, etc., in addition to other relevant types of information pertaining to the organisms (*Bradford et al., 2011*; *Bowes et al., 2010*; *Yook et al., 2012*; *Marygold et al., 2013*; *Laulederkind et al., 2013*; *Bult et al., 2013*). While it is reassuring that these specific strains have been previously curated via earlier publications, it often requires the curator to dig through many publications or to contact the authors directly. ZFIN determined that over a two-month period, they had to contact 29% of authors to properly curate the resources reported in their manuscript.

Comparing organism identification between disciplines, we noted that they were considerably less identifiable in Neuroscience papers (46%) relative to other domains. A likely explanation is that non-transgenic animals are commonly used in neuroscience assays such as electrophysiology studies (26 out of 62 organisms analyzed were non-transgenic). Identification of such commercially available strains faces similar problems as standard cell lines, where a source is required to allow some historical information to be obtained about

propagation/breeding. Indeed, it has been reported that there are many variations between wild type strains of model organisms (*Portelli et al., 2009*; *Sandberg et al., 2000*; *Wahlsten, 1987*), and variations between suppliers (*Ezerman & Kromer, 1985*).

## Domain considerations

We further examined if the unique identification of resources differed between sub-disciplines of biomedical research (Table 1). While no discipline was consistently above or below average with respect to identification of the resources, Developmental Biology, General Biology, and Immunology were generally above average compared to the other fields. The identification of cell lines was highest in Immunology papers, which was significantly different from Cell and General Biology papers, and papers from the "other" category, even though there was a small sample size (16 out of 104 total cell lines were from Immunology journals). By contrast, no cell lines were identifiable in the General Biology papers, which was significantly lower compared to all disciplines except the "other" category. However, General Biology journals boasted the highest percentage of identifiable constructs in papers at 59%, which was a significantly better compared to the other disciplines except Immunology. It is notable that identification of resources for Neuroscience was below average compared to the other fields for all resources except cell lines. Of note, identification of organisms in Neuroscience journals was significantly less than all other disciplines (30 out of 62 organisms were identifiable). Overall, there was not a consistent trend between scientific sub-domains with respect to identifiability of resources (Figs. 1B–1F).

## Impact factor considerations

We next examined whether identification of resources differed among journals across a range of impact factors. We found that resource identification did not vary with journal impact factor, as revealed by the lack of correlation in scatter plot analysis (Figs. 2A–2E).

## Analysis by reporting requirements

Very few journals were considered to have stringent reporting requirements, and amongst those, it was surprising to note that the identification of the resources did not appear improved in journals with satisfactory or loose reporting requirements. Identification of cell lines was especially poor in journals with satisfactory reporting guidelines (0 out of 21 were identifiable, from 10 articles analyzed), and overall, the identification of the resources was the poorest in journals with highest reporting requirements (an average of 45% were identifiable in journals with stringent reporting requirements, while resources from journals with satisfactory and loose were on average 61% and 55% identifiable, respectively; Fig. 3). On average, journals with loose reporting requirements had a significantly higher percentage of identifiable resources compared to journals with stringent reporting requirements.

With most journals having a low or mid-level impact factor (i.e., a skewed distribution), the majority of high identifiability therefore comes from these lower profile journals. This is an encouraging result, because it means that the lion's share of the publishing

**Table 1** Final numbers of identifiable resources in each domain.

| Resource type | Domain | Total number identifiable | Total number of resources | Total number of papers | Percentage identifiable |
|---|---|---|---|---|---|
| **Antibody** | Cell biology | 69 | 149 | 34 | 46% |
| | Dev biology | 68 | 144 | 44 | 47% |
| | General biology | 36 | 74 | 19 | 49% |
| | Immunology | 48 | 124 | 28 | 39% |
| | Neuroscience | 60 | 136 | 41 | 44% |
| | Other | 31 | 76 | 24 | 41% |
| | **Grand total** | **312** | **703** | **190** | **44%** |
| **Cell lines** | Cell biology | 15 | 38 | 17 | 39% |
| | Dev biology | 7 | 12 | 5 | 58% |
| | General biology | 0 | 10 | 5 | 0% |
| | Immunology | 14 | 16 | 6 | 88% |
| | Neuroscience | 4 | 7 | 6 | 57% |
| | Other | 5 | 21 | 9 | 24% |
| | **Grand total** | **45** | **104** | **48** | **43%** |
| **Constructs** | Cell biology | 16 | 84 | 17 | 19% |
| | Dev biology | 18 | 66 | 19 | 27% |
| | General biology | 16 | 27 | 8 | 59% |
| | Immunology | 3 | 8 | 3 | 38% |
| | Neuroscience | 4 | 35 | 7 | 11% |
| | Other | 7 | 38 | 12 | 18% |
| | **Grand total** | **64** | **258** | **66** | **25%** |
| **Knockdown reagents** | Cell biology | 40 | 49 | 16 | 82% |
| | Dev biology | 55 | 76 | 22 | 72% |
| | General biology | 31 | 31 | 9 | 100% |
| | Immunology | 5 | 5 | 3 | 100% |
| | Neuroscience | 9 | 12 | 6 | 75% |
| | Other | 35 | 37 | 14 | 95% |
| | **Grand total** | **175** | **210** | **70** | **83%** |
| **Organisms** | Cell biology | 57 | 70 | 27 | 81% |
| | Dev biology | 119 | 141 | 44 | 84% |
| | General biology | 30 | 36 | 11 | 83% |
| | Immunology | 38 | 48 | 20 | 79% |
| | Neuroscience | 30 | 62 | 38 | 48% |
| | Other | 57 | 71 | 28 | 80% |
| | **Grand total** | **331** | **428** | **168** | **77%** |
| Overall total | | 927 | 1703 | | 54% |

world has already demonstrated a capability of producing identifiable resources. It is especially important to not overlook these higher volume lower-cited journals to produce quality metadata about research resources. Additionally, higher impact journals tend to de-emphasize methods over other sections. Therefore, what is needed is to incentivize all journals to do better with respect to identifiability.

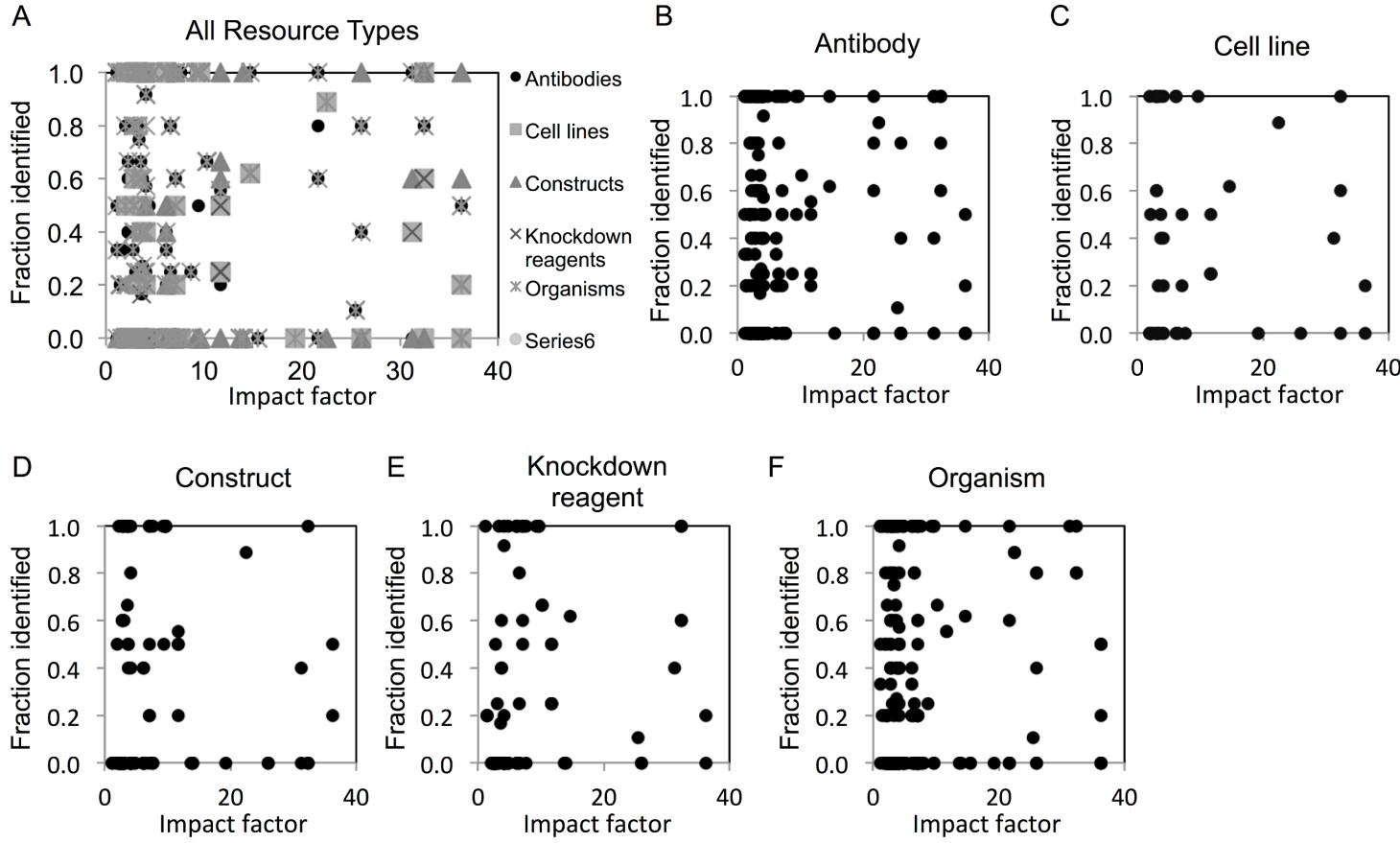

**Figure 2  Resource identification rates across journals of varying impact factors.** (A) An overview of fraction identified by impact factor for all resource types. (B–F) Fraction identified by impact factor for each individual resource type. Increasing height on the *x*-axis corresponds with a higher impact factor for each journal.

## Lab documentation vs. publications

For the Urban lab publications that we evaluated, only 44% of the antibodies used were identifiable (out of 9 total antibodies from 5 papers), and 47% of the organisms were identifiable (out of 17 organisms from 17 papers). We note that this lab internally keeps highly structured notes and metadata about their resources in the lab; after analyzing their internal notes, 100% of antibodies and 100% of organisms were identifiable using our criteria. However, despite this information being tracked extensively within the lab, these details did not make it into their publications. It does suggest, however, that the information is potentially recoverable, if practices to make resources identifiable are implemented.

## Evaluation criteria and workflow

A core challenge of designing this experiment was determining evaluation criteria that were precise enough to allow for reproducible determination of reported resource identifiability. For simplicity, we used a binary classification for the data analysis, but in

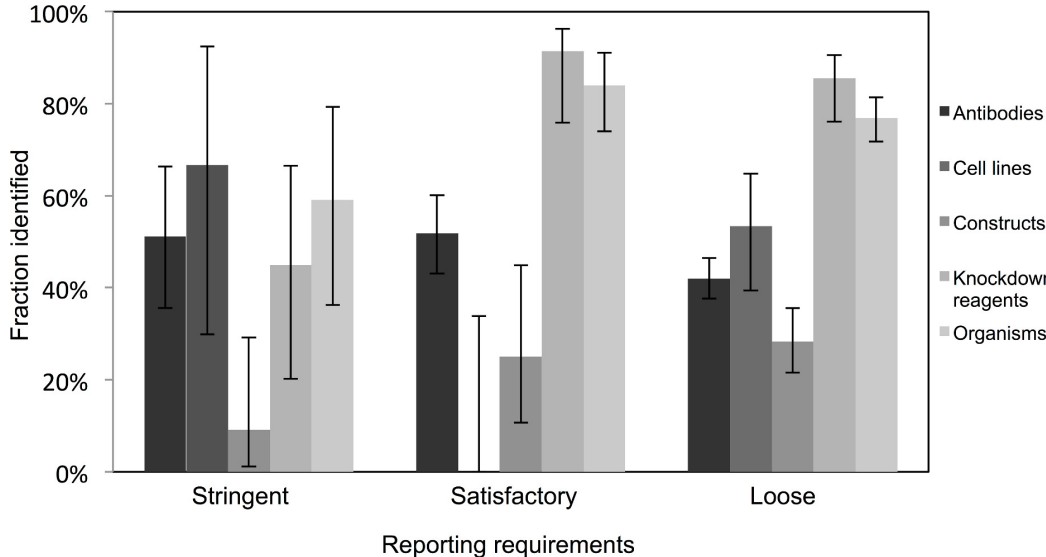

**Figure 3 Identification of resource varies across journals with varying resource-reporting requirements.** The classifications of reporting requirements are summarized in the methods. A total of 53 out of 118 resources were identifiable in the stringent reporting guidelines category (17 papers were analyzed), 201 resources were identifiable out of 329 resources for the satisfactory category (48 papers were analyzed) and 662 out of 1,217 resources were identifiable in the loose category (182 papers were analyzed). Variation from this average is shown by the bars, error bars indicate upper and lower 95% confidence intervals.

reality the amount of information pertaining to resource identification was incremental. Crafting of these criteria required careful consideration of each resource type, including how they are generated and acquired and the particular aspects of each that are important in the context of experimental reproducibility. This was particularly complex for resources whose identification required sequence information relating to a target or part of the resource, as different applications may require different degrees of specificity. Despite the abundance of public databases that provide identifiers for biological sequences, we found a reluctance of authors to reference such IDs when documenting reagents such as constructs or antibodies. This may point to a lack of awareness, a lack of incentive, or a lack of means for the journals and authors to use existing resources to supply uniquely identifiable information. Each problem is likely to have its own set of solutions, which we discuss in our recommendations below.

To ensure their consistent application, criteria and evaluation workflows were centrally documented, performed, and evaluated performed by expert biocurators. These results support the specificity and reproducibility of our guidelines, which we hope will serve to inform reporting requirements of publishers and the development of support platforms for authors.

## CONCLUSIONS

Improving reporting guidelines for authors is an important step towards addressing this problem. Very few journals (only 5/83) had high stringency guidelines by our

definition. Higher impact journals like Science and Nature tended to have looser reporting requirements, usually due to space limitations in the journal and often required reference to previously published methods. It has also been previously noted that higher impact journals have a higher retraction rate (*Fang, Casadevall & Morrison, 2011*). The Journal of Comparative Neurology has stringent reporting standards for materials and methods, requiring that sources for all materials and equipment, sequence information for nucleic acids and peptides, and immunogen and catalog number for antibodies be reported. It is our hope that other journals will follow suit. That said, we found that antibody identifiability in the Journal of Comparative Neurology was only slightly higher than average across all journals (58% in JCN vs. 44% overall). Our findings are also much lower than the percentage calculated from the JCN database above, perhaps due to lack of compliance by authors or lack of enforcement by reviewers. Based on the sampling that we have, there does not seem to be any relationship between reporting guidelines and identifiability. One might ask, how can this be? The reality is that having quality guidelines for authors is only one part of the solution. For example, Mike Taylor writes about how the peer review process fails to enable trustworthy science (*Taylor, 2013*).

The solution to improving resource identifiability and therefore scientific reproducibility needs to be a partnership between all participants in the scientific process, and deficiencies in awareness and difficulties coordinating across these stakeholders is at the root of the problem. Better tracking of research resources by researchers during the course of research can facilitate sharing of information with databases and at publication time. Electronic lab notebooks and management software (*Machina & Wild, 2013*; *Hrynaszkiewicz, 2012*), or resource sharing repositories such as the eagle-i Network (www.eagle-i.net) (*Vasilevsky et al., 2012*) or the Neuroscience Information Framework (http://www.neuinfo.org/) (*Bandrowski et al., 2012*) enable creation of stable identifiers and structured tracking of information. The MODs have recommended nomenclature standards for organisms, but these are not always adhered to (*Eppig & Levan, 2005*; *MGI, 2013*; *ZFIN, 2013*; *Flybase, 2013*). In an ideal situation, authors would report the unique ID pertaining to the model organism directly in the publication by having their ID assigned and nomenclature approved prior to publication. Then a direct link and easy access to the information to researchers who are attempting to understand or reproduce an experiment can be made available. In addition, this can facilitate text-mining and machine processing using automated agents that recognize these IDs. Journal editors should better detail reporting requirements, such as in the recent communiqué from Nature (http://www.nature.com/authors/policies/reporting.pdf). Publishers also need functionality to identify resources at the time of submission. Tools such as the DOMEO Toolkit allow for semantic markup of papers (*Ciccarese, Ocana, & Clark , 2012*) and can be utilized during the submission process whereby researchers can easily check the identifiability of the resources found in their paper. Vendors, if more aware of how their products are being referenced in the literature and databases, may tend towards better and more stable catalog schemes as well as to integrate the added knowledge being captured in external resources. Finally, researchers can be attributed for their resources so that they would be incentivized

to uniquely identify and share them. Recent changes to the NSF biosketch highlight a specific area where uniquely identifying such resources can have a positive influence on the evaluation of one's scholarly activities. Similarly, the Bioresource Research Impact Factor (BRIF) (*Mabile et al., 2013*) provides attribution for use and sharing of resources. Unique reference of resources through databases such as the Antibody Registry, eagle-i or MODs can facilitate this process. Finally, researchers need to know where the information in their favorite online resources comes from—the literature and the biocurators that curate their papers and datasets. Identifiability is just as important in the context of data sets, and given the significant effort being made to make informatics analyses reproducible (http://www. runmycode.org/CompanionSite/) and data sets available (dryad.org, figshare.com/), it is ironic that in some cases the original data itself may not be reproducible simply because the antibody used to generate the data was never specified.

Scientific reproducibility is dependent on many attributes of the scientific method. Being able to the uniquely identify the resources used in the experiments is only one of these attributes—it just happens to be the easiest one to accomplish. We hope that this study insights authors, reviewers, editors, vendors, and publishers to work together to realize this common goal.

## ACKNOWLEDGEMENTS

We would like to acknowledge Robin Champieux for her help with the experimental design, John Campbell for his help with data and statistical analysis and discussion, Scott Hoffmann for his help with the data analysis and figure preparation, Alex Hodgson for sharing the antibody market analysis and manuscript review, Nathan Urban for discussions and sharing information on lab internal databases and notes, and Randi Vita for manuscript review and for sharing the IEDB data, Ceri Van Slkye for her help with analyzing the yeast strains, and Anita de Waard, Maryann Martone, and Anita Bandrowski for discussion and manuscript review.

### Funding

OHSU acknowledges the support of the OHSU Library and #1R24OD011883-01 from the NIH Office of the Director. The Zebrafish Information Network and Flybase are funded by the National Human Genome Research Institute (P41 HG002659 and P41 HG000739, respectively). Shreejoy Tripathy of the Urban Lab is funded by an NSF graduate research fellowship and a RK Mellon Foundation Fellowship. Greg LaRoca is funded by NIH grants R01DC005798 and R01DC011184. The funders had no role in study design, data collection and analysis, decision to publish, or preparation of the manuscript.

### Grant Disclosures

The following grant information was disclosed by the authors:
NIH Office of the Director: NIH grant #1R24OD011883-01.
National Human Genome Research Institute: #P41 HG002659, #P41 HG000739.

NSF graduate research fellowship and a RK Mellon Foundation fellowship. NIH grants: R01DC005798, R01DC011184.

## Competing Interests

The authors have no competing interests.

## Author Contributions

- Nicole A. Vasilevsky conceived and designed the experiments, performed the experiments, analyzed the data, wrote the paper.
- Matthew H. Brush and Melissa A. Haendel conceived and designed the experiments, wrote the paper.
- Holly Paddock and Laura Ponting performed the experiments.
- Shreejoy J. Tripathy provided data from the Nathan Urban Lab that we used in the analysis and reviewed the manuscript.
- Gregory M. LaRocca provided data from Nathan Urban Lab that we used in the analysis.

## Data Deposition

The following information was supplied regarding the deposition of related data:
figshare: http://dx.doi.org/10.6084/m9.figshare.773119.

## Supplemental Information

Supplemental information for this article can be found online at http://dx.doi.org/10.7717/peerj.148.

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
