# Peer review of "On the reproducibility of science: unique identification of research resources in the biomedical literature"

_PeerJ, doi:10.7717/peerj.148_

## Round 0.1 · original submission · Major Revisions

Dear Authors,I hope all of you can contribute to the necessary revisions needed so that the same peer reviewers can re-review them.

Reviewer 1 ·

Basic reporting

The manuscript reports findings on a very important problem in biomedical literature and will serve to be very useful in many dimensions.

Experimental design

The authors have carefully chosen 5 areas or reagents that are often not reported accurately in the literature. Here are some experiments that would add value or strengthen the claim of the paper.
1) In papers where the reagent was not easy identifiable, it would be interesting to know how many other papers have cited this paper and claimed to have prepared their reagents using the methods used by that paper.
2) The authors don't say much about whether there is a bias in the number of papers they selected. Is 135+86+17 a good representation, is that enough?How many of these papers had no relevance to this study, i.e were not experimental papers requiring reporting of constructs, antibodies etc.
3) Would it have been possible to randomly ask a subset of authors from papers that don't have indentifiable resources to see why they did not report the details. Is it because the journals don't have a structured form to report these details? From the data presented, it almost seems like the stringency has no bearing on the # of identified resources.
4) Is it possible that there are way too many reagents to report for any given paper and authors report the resources for just a select set of reagents that are most relevant to understanding the point of the paper?
4) In terms of recommendations, the authors seems to suggest a lot of different options, but none of them are very decisive. So far journals have been successful in requiring submission of sequence to GenBank or structure coordinates to PDB and authors for the most part stick to this requirement irrespective of the name of the journal. How has this been successful and is there something to learn and extend for other resources?

Validity of the findings

The findings reported in this paper are valid.

·

Basic reporting

No Comments

Experimental design

No Comments

Validity of the findings

1. p.6, on the discussion of the difficulty of identifying cDNA or peptides related to a gene: have the authors also encountered issues with identifying the species in which the sequence was isolated ? There have been reports that this is problematic.
2. I am a bit surprised about the identification of organisms: usually yeast, frogs, worms, and flies are relatively easy to unambiguously identify; moreover, there is no data for human. How was the analysis done ? For example, 0 % of the yeast were identified, but is there evidence that it should have been the case, in other words, were there any yeasts papers in the set ?

Additional comments

This paper addresses the very pertinent issue of the lack of sufficient information provided in papers to allow research to be accurately reproduced. The article is very well written, and the results are interesting and provide some quantitative measure of the extent of the problem.

Minor comments:
1. p. 7, "Statical analysis": the section title should be marked with underline the same way as 'Journal selection and classification' above.
2. p. 9, Section on Cell lines: There seems to be missing something in the sentence "A source for cell lines was rarely reported and was most common factor for their low identifiability in our study"; please rephrase.
3. At the end of the same paragraph the authors write "see methods section"; it is not clear what they refer to, since this is already the methods section.
4. p. 11: "While it is assuring" -> "While it is reassuring"

---

## Round 0.2 · accepted · Accept

Congratulations for the manuscript being accepted. Do send more high quality manuscripts to PeerJ again.

·

Basic reporting

NA

Experimental design

NA

Validity of the findings

NA

Additional comments

All my comments have been addressed.

Reviewer 2 ·

Basic reporting

The authors have addressed my concerns.

Experimental design

No comments

Validity of the findings

No Comments

---

## Author Rebuttal · Round 0.2

# Reviewer Comments

**Reviewer 1**
**Basic reporting**
The manuscript reports findings on a very important problem in biomedical literature and will serve to be very useful in many dimensions.

**Experimental design**
The authors have carefully chosen 5 areas or reagents that are often not reported accurately in the literature. Here are some experiments that would add value or strengthen the claim of the paper.

1) In papers where the reagent was not easy identifiable, it would be interesting to know how many other papers have cited this paper and claimed to have prepared their reagents using the methods used by that paper.

Authors response:  One of the issues in performing a citation tracking study of this nature (as we understood the request), is that we'd need to have performed this analysis on older papers. In this study, the papers chosen were all newly published papers, to avoid having had the resources already tracked down and curated by the various databases. In this study, we did go back and review any prior publications that were referenced in the paper being analyzed, as well as any referenced in the prior paper, and so on. Furthermore, most authors don't reference previous citations for commercial resources.

Based on our collective experience in this study and in curation of the literature in general, it is usually the case that previous articles still do not contain sufficient information to identify the resource and the authors must be contacted. For example, in the manuscript we state that "ZFIN determined that over a two-month period, they had to contact 29% of authors to properly curate the resources reported in their manuscript" - this included the review of several publications in the chain of reference.

2) The authors don't say much about whether there is a bias in the number of papers they selected. Is 135+86+17 a good representation, is that enough? How many of these papers had no relevance to this study, i.e were not experimental papers requiring reporting of constructs, antibodies etc.

- NV: I am not really sure I understand what they are asking here
- HP: perhaps they think some of the papers were reviews? Which was not the case, though some of them may have been looking at behavior or something else that is non reagent intensive. Also it appears that they think the number is rather low.

Author response: We are not entirely sure what the reviewers are requesting, but have added some clarifying text to the methods on page 5.  We do not believe that any of the papers we analyzed had no relevance to the study, as all were chosen based on their inclusion of relevant resources and we excluded reviews or summary papers.

3) Would it have been possible to randomly ask a subset of authors from papers that don't have identifiable resources to see why they did not report the details. Is it because the journals don't have a structured form to report these details? From the data presented, it almost seems like the stringency has no bearing on the # of identified resources.

Author response: This is an important point and is the reason for the inclusion of Urban lab analysis - to gain some insight from an organized lab about how they go about citing research resources.

This is a very good idea, and prospectively we will work on collecting this data; however, a survey of this nature will require IRB approval and is therefore not in scope for this manuscript.

4) Is it possible that there are way too many reagents to report for any given paper and authors report the resources for just a select set of reagents that are most relevant to understanding the point of the paper?

Author response: While it is true that in some cases we limited our analysis to 5 resource types of the same category in each paper (which is stated in the methods section and highlighted on page 5), it is also the case that authors are required to report all research resources used in the course of their study.

5) In terms of recommendations, the authors seems to suggest a lot of different options, but none of them are very decisive. So far journals have been successful in requiring submission of sequence to GenBank or structure coordinates to PDB and authors for the most part stick to this requirement irrespective of the name of the journal. How has this been successful and is there something to learn and extend for other resources?

Author response: In the manuscript, we have included lengthy guidelines for how we determined "identifiability". Theoretically these could be used by journals as well, however, we fully recognize that we necessarily had to be very rigorous so as to enable rigorous downstream statistical analysis. We have therefore provided a more suitable, streamlined version of these guidelines as a new data standard for resource identification and have posted them on the Force11 (http://www.force11.org/node/4433), and Biosharing (http://biosharing.org/bsg-000532) websites. We are also now working with publishers to facilitate their inclusion in guidelines to authors.

It should be noted, however, that even in the case of journals with stringent guidelines, that identifiability was not improved. We therefore suggest that reviewers be made more aware of such requirements as part of their review process, and ideally, we also urge publishers to include tools that can facilitate identification of research resources into the submission and review process.

As for Genbank, in the 1990s, journals started requiring accession numbers for sequence data, and urged authors and reviewers to consider this. If you look at most journal guidelines today, they don't actually include this in the guidelines specifically. It has just become common practice to include them - though in our experience, there are still plenty of authors that do not.

**Validity of the findings**

The findings reported in this paper are valid.

**Reviewer 2 (Pascale Gaudet)**

**Basic reporting**

No Comments

**Experimental design**

No Comments

**Validity of the findings**

1. p.6, on the discussion of the difficulty of identifying cDNA or peptides related to a gene: have the authors also encountered issues with identifying the species in which the sequence was isolated ? There have been reports that this is problematic.

Author response: In our data set, we were unable to identify genes (and corresponding source organisms) for 75% of constructs and 17% of knockdown reagents. However, in our experience, it is often the case that authors do not include the species of the sequence in question, but if the sequence is provided, blasting is always an option.

2. I am a bit surprised about the identification of organisms: usually yeast, frogs, worms, and flies are relatively easy to unambiguously identify; moreover, there is no data for human. How was the analysis done ? For example, 0 % of the yeast were identified, but is there evidence that it should have been the case, in other words, were there any yeasts papers in the set ?

Author response: We re-reviewed the data for yeast, frogs and worms. There were only 5 yeast strains from one paper that were analyzed in this study. We consulted with a yeast expert and found that these strains were in fact identifiable and reanalyzed the data and adjusted the results accordingly. The numbers of the frog and worm strains were also low (3 and 11 strains total, respectively) and we confirmed the numbers were accurate. There are no human organisms, because there is no mechanism to look up "strains of human" in any model organism database. We therefore did not include humans as a model organism type for our analysis. In such a case, one would want to have a dbGAP identifier or other anonymized identifier, or family or population identifier. These are also rarely obtained or reported upon.

**Comments for the author**

This paper addresses the very pertinent issue of the lack of sufficient information provided in papers to allow research to be accurately reproduced. The article is very well written, and the results are interesting and provide some quantitative measure of the extent of the problem.

Minor comments:

1. p. 7, "Statical analysis": the section title should be marked with underline the same way as

'Journal selection and classification' above.
- Done

2. p. 9, Section on Cell lines: There seems to be missing something in the sentence "A source for cell lines was rarely reported and was most common factor for their low identifiability in our study"; please rephrase.
- Rephrased to "A source for cell lines was rarely reported and the lack of source was most common factor for their low identifiability in our study"

3. At the end of the same paragraph the authors write "see methods section"; it is not clear what they refer to, since this is already the methods section.
- This paragraph is in the Results and Discussion section

4. p. 11: "While it is assuring" -> "While it is reassuring"
- This has been changed to "reassuring"